# Deep blind arterial input function: signal correction in perfusion cardiac magnetic resonance

**Habib Rebbah**[1]                                           HABIB.REBBAH@OLEA-MEDICAL.COM
**Magalie Viallon**[2,3]                              MAGALIE.VIALLON@CREATIS.INSA-LYON.FR
**Pierre Croisille**[2,3]                             PIERRE.CROISILLE@CREATIS.INSA-LYON.FR
**Timothé Boutelier**[1]                                 TIMOTHE.BOUTELIER@OLEA-MEDICAL.COM

[1] *Research and Innovation department, Olea Medical, La Ciotat, France*

[2] *Univ Lyon, UJM-Saint-Etienne, INSA, CNRS UMR 5520, INSERM U1206, CREATIS, F-42023, Saint-Etienne, France*

[3] *Department of Radiology, University Hospital Saint-Etienne, Saint-Etienne, France*

**Editors:** Accepted for publication at MIDL 2024

## Abstract

*Objectives*: The non-linear relationship between gadolinium concentration and the signal in perfusion cardiac magnetic resonance (CMR) poses a significant challenge for accurate quantification of pharmacokinetic parameters. This phenomenon primarily impacts the arterial input function (AIF), causing it to appear saturated in comparison to the temporal concentration profile. This study aims to leverage a blind deconvolution strategy through a deep-learning approach to address the saturation in the AIF.

*Methods*: We propose the utilization of a convolutional neural network (CNN) architecture with the saturated AIF and a set of myocardial tissue signals as inputs, generating the corrected AIF as the output. To train the network, a dataset comprising over $3 \times 10^6$ simulated AIFs with associated signals from five simulated tissues response for each instance was employed. To assess the effectiveness of the approach, the trained network was evaluated using a dual-saturation sequence to compare the corrected AIF with the unsaturated version. The clinical dataset encompassed scans from 43 patients.

*Results*: The mean square error (MSE) for the testing subset of the simulated database was 0.69% of the peak. In the in vivo dataset, the coefficient of determination R2 was 0.26 and 0.86 for the saturated and corrected AIF, respectively, in comparison to the unsaturated AIF.

*Conclusion*: The proposed network successfully corrects the acquisition-induced effects on the AIF. Moreover, the extensive simulated database, featuring diverse acquisition parameters, facilitates the robust generalization of the network's application.

**Keywords:** Perfusion, MRI, AIF, CNN.

## 1. Introduction

The intravenous administration of gadolinium-based contrast agent (CA) in cardiovascular magnetic resonance (CMR) during its initial vascular transit has become a clinical standard for assessing myocardial perfusion states (Jerosch-Herold et al., 2006). Specifically, dynamic contrast enhanced (DCE) imaging in the cardiac domain has demonstrated its efficacy in evaluating various diseases (Montalescot et al., 2013; Nagel et al., 2019). Perfusion quantification typically involves employing a deconvolution approach based on a parametric tracer-kinetic model and an arterial input function (AIF) (Sourbron and Buckley, 2011;

Daviller et al., 2021). The AIF is commonly derived from time-signal profiles within a region of interest (ROI) placed in the left ventricular (LV) blood pool.

However, the signal obtained is not linearly dependent on the concentration of the CA. Such a linearity can only be assumed for the lowest CA concentrations, which, in practice, can be expected for myocardial signals. The cardiac image acquisition sequence introduces effects often described as saturation. Typically, common sequences involve a saturation preparation followed by multiple acquisition pulses of spoiled gradient-recalled echo (SPGR) (Barkhausen et al., 2004; Slavin et al., 2001). The long recovery time ($\sim$90-110ms) used in CMR perfusion sequence to improve the contrast between remote to ischemic region in first-pass imaging lead non-linear attenuation of the AIF. Relying on signals from the LV blood pool could consequently lead to the misestimation of pharmacokinetic (PK) parameters. The signals acquired in such sequences can be theoretically described by acquisition and tissue parameters, including T1, T2, repetition time (TR), and flip angle (Hänicke et al., 1990; Hsu et al., 2008).

One can utilize a derived equation for the inverse problem to estimate the concentration profile from the LV blood pool signal (Hsu et al., 2008). However, this approach relies on a detailed description of the sequence used for acquisition, which may not always be available. Additionally, certain model limitations, such as the water exchange effect, can complicate the methodology (Landis et al., 2000).

Various solutions have been proposed to address these limitations. One popular approach involves using a population-based Arterial Input Function (AIF). This entails deriving a constant AIF by averaging AIFs from a population of volunteers or patients who share characteristics similar to those of the current patient under analysis (Parker et al., 2006). However, this method has the obvious drawback of not capturing individual variations, which can be substantial.

Some authors have suggested modifying the acquisition scheme or sequence. The dual-bolus method involves two injections: one, dedicated to the AIF acquisition, with a lower dose of contrast agent (CA) to limit saturation effects and another with a higher dose to maximize tissue response (Christian et al., 2004). However, the dual-bolus protocol remains complicated to implement for clinical routine with extensive care to be taken to ensure the reproducibility of the boluses, especially during stress.

The dual-saturation acquisition proposes obtaining two types of images during the same breath-hold (Gatehouse et al., 2004; Kim and Axel, 2006). The first image, with a short recovery time ($\sim$10ms), aims to limit the acquisition's impact on the signal, promoting linearity in the signal-concentration relationship but with low resolution. The second image follows common parameters of a perfusion sequence in CMR. The dual-imaging approach is still proposed as prototype sequence by most vendors. Hence, this solution is limited to research areas, and its clinical routine application requires specific training. Both dual-bolus and dual-saturation methods are constrained to their own acquisitions and cannot be generalized to the extensive dataset of perfusion acquisitions over time. Most of the perfusion acquired data in routine clinical practices are still obtained without satisfying solution and considering the potential pitfalls of using various elaborated perfusion pulse sequences in research studies, an attenuation resilient AIF estimation method based on conventional perfusion imaging protocols with high-dosage gadolinium boluses is of high interest.

Other research has focused on estimating the correct AIF directly from acquired images using a blind deconvolution approach (Schabel et al., 2010). This assumes a shared AIF among various tissues with different tissular responses, with the redundancy of information allowing simultaneous estimation of tissular pharmacokinetic parameters and the corresponding AIF. However, the practical application of such a method requires a sufficiently diverse range of tissue responses to ensure stability. One frequently suggested approach is to constrain the AIF to a specific parametric time-course model, such as Parker's AIF. Bayesian approaches have shown also promising results (Schmid et al., 2007). Despite the stability limits, these methods can be computationally intensive due to the conventional iterative resolution of the deconvolution problem for tissue response and AIF.

To overcome these limitations, recent studies propose using a deep learning approach to correctly estimate the AIF from the acquisition. One approach involves using unsaturated AIFs from dynamic susceptibility contrast (DSC) acquisitions and their corresponding DCE profiles to train a conditional generative adversarial network for correcting DCE saturated AIFs (Choi et al., 2020). Another approach uses a similar method based on unsaturated AIFs obtained using the dual-saturation sequence to train a 1D U-Net for correcting input DCE AIFs (Scannell et al., 2023). Although both methods present promising results, their generalization is limited by the variability in their databases, primarily due to differences in magnetic field strength (1.5 vs. 3T) and MRI machine characteristics.

To the best of our knowledge, the latest proposition introduces a form of blind deconvolution into a physics-informed neural network (PINN) to constrain AIF estimation (van Herten et al., 2022). Despite the extensive dataset and the absence of the need for labeled data, the main limitation is the considerable computation time required for this approach (1 hour per slice for a standard personal computer).

In this study, we propose a deep learning solution to estimate the AIF while maximizing generalization, based on simulations and blind deconvolution principle, taking saturated AIF and tissue signals as inputs to return the estimation of the unsaturated AIF. The objective is to develop a user-friendly and efficient solution that circumvents the need for complex modeling of acquisition sequences and can be extended to diverse acquisition parameter settings.

## 2. Methods

### 2.1. Simulated database

A large simulated database of AIFs and their corresponding tissue responses of over $3 \times 10^6$ cases is used to train the network. We adopted Parker's simplified form to generate the unsaturated AIFs, as described in (Parker et al., 2006).

$$\text{AIF}_{unsat} = A_0 \frac{\exp(-kt)}{1 + \exp(-s(t - \tau))} + \sum_{n=1}^{2} A_n \exp\left(-\frac{t - T_n}{2\theta_n}\right) \tag{1}$$

Five different tissue responses were generated for each AIF using the Toft-Kety model (Tofts and Kermode, 1991; Tofts et al., 1999). The concentration observed for each case can be computed as follows:

$$C_{tissue}(t) = \text{AIF}(t) \star R(t)$$
$$R(t) = K^{trans} \times \exp\left(-k_{ep}t\right) \tag{2}$$

A uniform, independent, and identically distributed random sampling of the parameters in Equation (1) and Equation (2) enabled the generation of 3,906,250 unique cases, with the upper and lower bounds described in Annex A. Eventually, the simulated AIF is converted into a saturated signal using a simplified set of Bloch equations tailored for cardiac perfusion acquisition, involving perfect saturation followed by SPGR pulses (equation 1.2.7 in (Rebbah, 2019)). Annex B provides details on the conversion process. Tissue concentrations were converted by assuming a linear approximation with their corresponding signals, employing a proportionality coefficient set arbitrarily to 1.

Due to the simulation aspect, time step units were used rather than absolute time length, employing a signal length of 40 steps for both AIF and tissue responses. Additionally, to avoid any scaling issues, we normalized all inputs and outputs—accurate AIF, saturated AIF, and tissue signals—to the $[0, 1]$ range.

## 2.2. In vivo database

The evaluation of the network was conducted using in vivo data obtained from a set of 43 patients/volunteers with dual-saturation acquisitions extracted from the in vivo database described in (Daviller et al., 2021). The perfusion sequences were acquired on a 3T MAGNETOM Prisma (Siemens Healthineers, Erlangen, Germany) at 3 to 5 short-axis locations for every heartbeat with a bolus injection (6 ml/s) of gadoterate meglumine (0.2 mmol/kg) (Dotarem, Guerbet, Paris, France).

The unsaturated AIF was computed by avering blood time curves derived from ROIs positionned in the cavity on the short recovery time acquisition series of the dual perfusion sequence acquisition. For the saturated acquisitions, an automatic registration was provided by the manufacturer (MOCO series, Siemens Healthcare). LV blood pool time curves, and myocardial time curves were obtained from ROIs manually delineated at the optimal contrast perfusion image to obtain the saturated AIF for correction and a set of myocardial signals. A linear interpolation is used if necessary to match inputs length of the proposed model. A K-means algorithm was employed to select 5 main signals, computed as the average signal of each cluster. To match the simulated signal length, all time profiles were linearly interpolated to obtain signals with a length of 40 time steps.

## 2.3. Network architechture and training

The proposed network is illustrated in Figure 1. It consists of a two-branch Convolutional Neural Network (CNN): a 1D branch for the saturated Arterial Input Function (AIF) and a 2D branch for the 5 tissue signals. The outputs of the two branches are flattened, combined, and processed through dense layers. The final output is a correction of the input saturated AIF. Annex D provides a comparison with a fully blind deconvolution model (without saturated AIF input).

The randomly allocated subdivision of the simulated dataset between training and validation follows an 80/20% ratio, corresponding to 3,198,093/781,250 cases. Training was

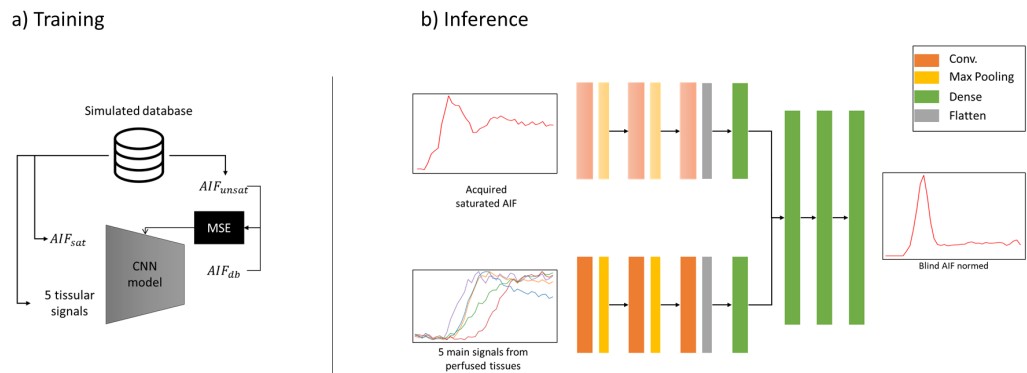

Figure 1: Deep blind AIF: CNN model. the model consists in two branches of CNN: 1D for the saturated AIF and 2D for the 5 main myocardial signals. The output is the estimated unsaturated AIF. Inputs and outputs are normalized in $[0, 1]$.

conducted using the ADAM optimizer with a learning rate of 0.001 and Mean Squared Error (MSE) loss between the proposed corrected AIF and the unsaturated simulated one. This training process spanned 30 epochs, with each epoch consisting of 20,000 cases per batch. The study was implemented using Python and the TensorFlow 2 framework.

The model is completed with a scale correction process. The output of blind deconvolution typically yields an estimate of the AIF that is scalefree, as all multiples of the proposed AIF satisfy the deconvolution problem, allowing the flow or $K^{trans}$ to absorb the variation. To estimate the correct scale factor, we followed recommendations in the literature and chose to match the area under the curve (AUC) of the tail of the correct AIF with the AUC of the tail of the saturated one (Schabel et al., 2010). Specifically, we selected the last 10 time points to represent the tail of the AIF.

## 3. Results

### 3.1. Training results

After 30 epochs, the model converged with a MSE of $0.69 \times 10^{-2}$ for both the training and validation subsets. Figure 2 showcases a few randomly selected results from the validation dataset. The unsaturated AIF (green line) is barely discernible, hidden by the prediction of the proposed algorithm (blue line).

However, we observed a peculiar phenomenon where, in each training iteration, certain temporal positions of the corrected AIF consistently exhibited values of zero. These specific temporal positions varied with each new training process but remained constant within the same iteration, independent of the input conditions. The cause of this behavior is unclear, but a straightforward solution was implemented. Following a successful training, the temporal positions where the corrected AIF exhibited zeros were identified and a post-processing step to interpolate the values at those time points was added. In the algorithm presented here, there were two instances of zero values. Refer to Annex E for further details.

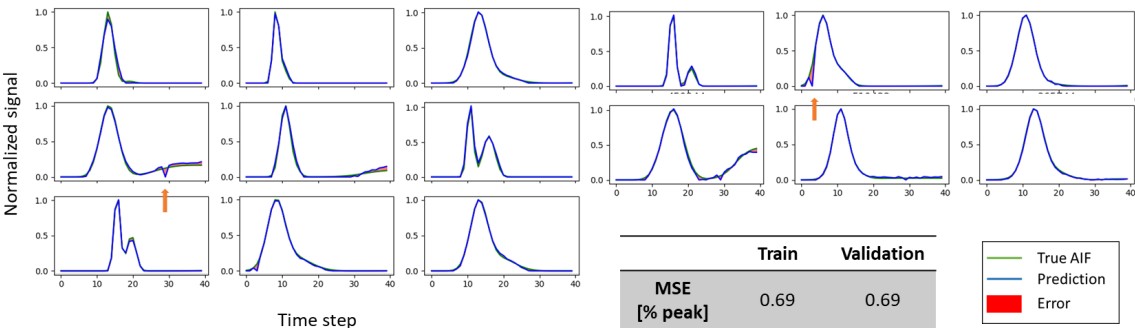

Figure 2: Training results. Few randomly selected examples from the validation dataset. The green line representing the unsaturated AIF when it is not apparent, is under the blue line, representing an accurate prediction. The orange arrows highlight the zeros points.

## 3.2. In vivo results

The comprehensive evaluation of the entire algorithm (comprising the network and the post-processing correction for scale and zero values) using dual-saturation sequences was conducted by comparing discrepancies in the unsaturated AIF ($AIF_{unsat}$) with the saturated one ($AIF_{sat}$) and the discrepancies of the $AIF_{unsat}$ and the corrected AIF estimated by our algorithm ($AIF_{db}$ for deep blind). The discrepancies were evaluated as MSE and coefficient of determination R2. Moreover, myocardial blood flow (MBF) was estimated using a deconvolution algorithm with a free-form residue function (Olea Medical), for each AIF along with the extracted main myocardial signals.

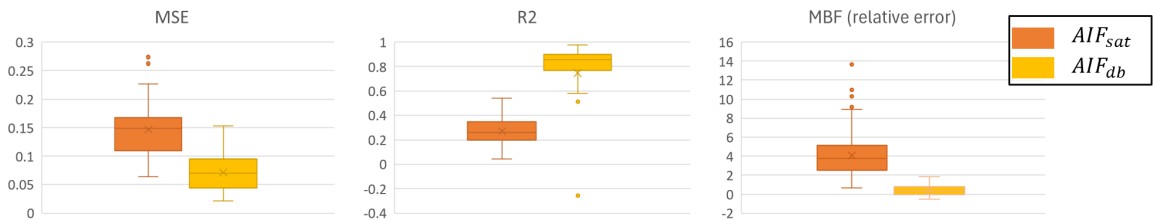

Figure 3: In vivo results. To compare the accuracy of the saturated $AIF_{sat}$ and the correction proposed in this study $AIF_{db}$. The accuracy is tested against the $AIF_{unsat}$ in case of dual-saturation sequences, in term of MSE and coefficient of determination R2. The upper right plot presents the relative error (in comparison to $AIF_{sat}$) of the estimated MBF using the $AIF_{db}$ against the $AIF_{unsat}$.

The results presented in Figure 3 and Figure 4 showcase some selected cases. The median MSE for the $\text{AIF}_{sat}$ was 0.15 +/- 0.05 compared to 0.07 +/- 0.03 for the $\text{AIF}_{db}$. The median R2 was 0.27 +/- 0.12 for the $\text{AIF}_{sat}$ compared to 0.86 +/- 0.45 (+/- 0.10 with outlier exclusion for two cases) of the $\text{AIF}_{db}$. The median relative error of MBF estimations was 3.78 +/- 1.99 for the $\text{AIF}_{sat}$ compared to the $\text{AIF}_{db}$ was 0.34 +/- 0.57. More in vivo results are gathered in Annex C.

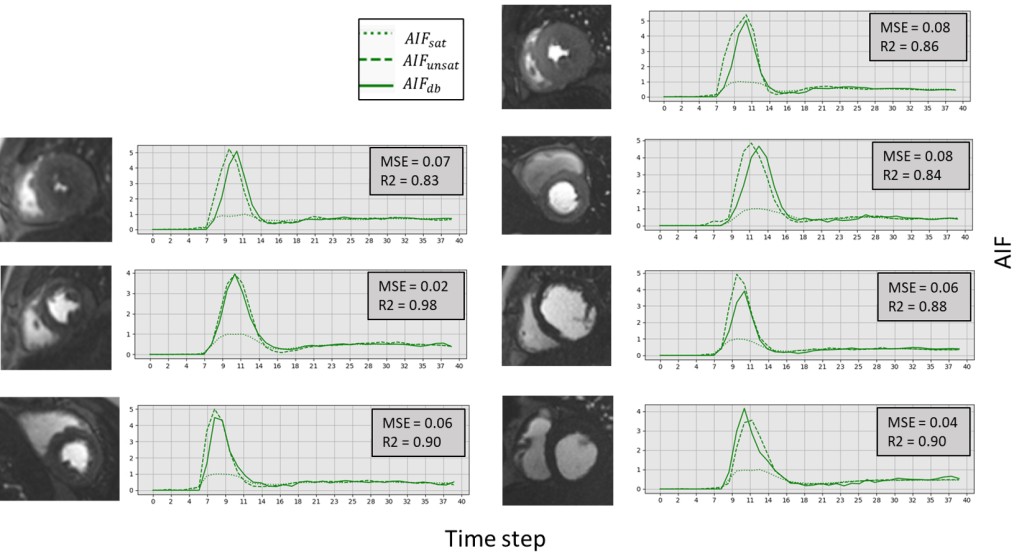

Figure 4: Selected in vivo results. Each case is represented by a cropped image around the heart with its associated AIFs at the right. For each case the MSE and R2 of the $\text{AIF}_{db}$ against the $\text{AIF}_{unsat}$ is provided. The AIFs are scaled such as the peak value of $\text{AIF}_{sat}$ is 1.

## 4. Discussion

The proposed approach demonstrated its capability to estimate the unsaturated version of the AIF accurately. Training the model with simulated data allows to easily explore perfusion parameters, enhancing its robustness and generalizability compared to strategies using limited number of in vivo data to train the network (Annex D). Figure 2 highlights that some simulated AIFs differ from real acquired AIFs. However, this variability promotes learning the underlying convolution operation between the AIF and tissue responses to derive tissue signals.

While the principle of deconvolution in perfusion imaging traditionally involves only tissue signals, our approach deviates from this by incorporating the saturated AIF, thereby constraining the deconvolution. In conventional literature (Schabel et al., 2010), this is achieved by choosing a parametric time-course model; however, our approach constrains the solution through the effect of the acquisition. The challenge then becomes finding

an AIF that can explain both the tissue signals and the transformation induced by the acquisition leading to the observed saturated AIF. Previous research suggested that ignoring tissue signals and using only the saturated AIF could accurately predict its unsaturated version (Choi et al., 2020; Scannell et al., 2023). However, we argue that such results primarily reflect the low variability of the datasets used for training in those studies, as they utilized perfusion acquisitions from very few centers with limited variations in acquisition parameters. Incorporating both the saturated AIF and tissue signals in our approach allows for better constraint of the solution. Moreover, choosing simulation with high parameter variations supports the validity of the results.

The solution presented here is preliminary work intended to demonstrate the network's ability to correct acquisition effects. However, the model proposed generates fixed zeros whose origin cannot be currently explained. Preliminary tests with the addition of multiple dropouts seem to address this issue effectively (Annex E), warranting further investigation. Nevertheless, the proposed post-processing successfully passed the evaluation test with real acquisition data. Notably, we did not introduce noise to the training data. This choice is derived from the practical pipeline induced by our method. Indeed, all inputs are averaged from a group of signals, using ROIs of the LV blood pool for the saturated AIF and myocardial signal classification for the five main tissue signals. This process significantly enhances the signal-to-noise ratio. Future work will focus on refining this aspect to enable the use of direct signals. However, the successful evaluation on real acquired data supports our initial intuition.

An intriguing open question concerns the performance dependency on input tissue signals. Specifically, the decision to solely utilize myocardial signals for deriving the five principal signals warrants consideration. Notably, a homogeneous myocardium may likely decrease the algorithm's performances, as evidenced by the outlier results observed in vivo. Furthermore, the adoption of K-means clustering to extract the main signals as cluster centroids is noteworthy. If the chosen number of clusters is restricted relative to the variability of the acquired signals, erroneous main signals may be generated through the averaging of disparate signals within the same cluster. Subsequent efforts will predominantly concentrate on precisely quantifying thus constraints and exercising greater discretion in selecting inputs.

## 5. Conclusion

A CNN model was proposed to correct the impact of acquisition on the signal to concentration relationship in perfusion CMR. The suggested model simulates blind deconvolution while imposing constraints on the desired AIF based on the concentration-to-signal conversion derived from simplified Bloch equations. The proposed strategy only needs a saturated AIF and myocardial signals to estimate an unsaturated AIF. Training the algorithm with simulated data allows testing a wide range of acquisition parameters. The evaluation against AIF derived from short recovery time acquisitions of dual-saturation acquisition demonstrates the validity of the approach.

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

## Appendix A. Simulations description

### A.1. AIF simulation

The AIFs were generated using a fixed time length (94s) with 40 evenly spaced samples. The equation 1 was rewrite in the following form:

$$\text{AIF}_{unsat}(t) = p_0 \exp\left(-\frac{t - t_{max}(p_3 + .1)}{2p_6}\right) + p_1(1-p_2)\exp\left(-\frac{t - (p_3 + t_{max}(1 + p_4)/10)}{2p_6 p_7}\right) +$$
$$p_1 p_2 \frac{\exp(-t/p_8)}{1 + \exp\left(-(t - t_{max}(p_4 + (1 + p_4 p_5)/10))/p_9\right)}$$

with the following values:

$$p_0 = 1$$
$$p_1 \in \{x \in \mathbb{R} \mid 0.1 \leq x \leq 0.75, \Delta x = 0.13\}$$
$$p_2 \in \{x \in \mathbb{R} \mid 0.1 \leq x \leq 0.9, \Delta x = 0.16\}$$
$$p_3 \in \{x \in \mathbb{R} \mid 0.05 \leq x \leq 0.3, \Delta x = 0.05\}$$
$$p_4 \in \{x \in \mathbb{R} \mid 0.05 \leq x \leq 1.5, \Delta x = 0.15\}$$
$$p_5 \in \{x \in \mathbb{R} \mid 0.75 \leq x \leq 1, \Delta x = 0.05\}$$
$$p_6 \in \{x \in \mathbb{R} \mid 2 \leq x \leq 7, \Delta x = 1\} \times 10^3$$
$$p_7 \in \{x \in \mathbb{R} \mid 1 \leq x \leq 2, \Delta x = 0.2\}$$
$$p_8 \in \{x \in \mathbb{R} \mid 1000 \leq x \leq 10^6, \Delta x = 10^6/5\}$$
$$p_9 \in \{x \in \mathbb{R} \mid 100 \leq x \leq 10^4, \Delta x = 10^4/5\}$$

Generating AIFs from all the combinations leads to a dataset of 3,906,250 unique cases.

### A.2. Delay simulation

As described in (Schabel et al., 2010), we simulated the delay using a convolution with a gaussian by randomly sampling the following parameters:

$$\text{disperssion} = 0.5$$
$$\text{delay} \in \{x \in \mathbb{R} \mid 0 \leq x \leq 10, \Delta x = 2\}$$

### A.3. Tissue signal simulation

We simulated the tissues signals using the Toft-Ketty model, by randomly sampling the following parameters:

$$k_{ep} \in \{x \in \mathbb{R} \mid 1 \leq x \leq 5, \Delta x = 1\} \times 10^{-2}$$
$$K^{trans} \in \{x \in \mathbb{R} \mid 1 \leq x \leq 3, \Delta x = 0.4\} \times 10^{-2}$$

## A.4. Conversion simulation

Using the equation 3 in Annex B, we simulated the conversion of $\text{AIF}_{unsat}$ to $\text{AIF}_{sat}$ by randomly sampling the following parameters:

$$M_0 = 1$$
$$T1 \in \{x \in \mathbb{R} \mid 1400 \le x \le 2000, \Delta x = 120\}$$
$$T2 \in \{x \in \mathbb{R} \mid 200 \le x \le 300, \Delta x = 1\}$$
$$r1 \in \{x \in \mathbb{R} \mid 3 \le x \le 6, \Delta x = 3/5\} \times 10^{-3}$$
$$r2 \in \{x \in \mathbb{R} \mid 4 \le x \le 7, \Delta x = 3/5\} \times 10^{-3}$$
$$TI \in \{x \in \mathbb{R} \mid 80 \le x \le 200, \Delta x = 24\}$$
$$TR \in \{x \in \mathbb{R} \mid 1 \le x \le 3, \Delta x = 2/5\}$$
$$TE \in \{x \in \mathbb{R} \mid 1 \le x \le 3, \Delta x = 2/5\}$$
$$\alpha \in \{x \in \mathbb{R} \mid 10 \le x \le 35, \Delta x = 5\}$$
$$N \in \{64, 65, 66, 128, 129, 130\}$$

## Appendix B. Conversion concentration to signal

To convert concentration to signal for generating saturated AIF, the following equation was used:

$$S([C]) = P_{T2} \times [(M_{z,1} - P_{eq}) \times P_{T1} + N \times P_{eq}] \qquad (3)$$
$$P_{T2} = \sin(\alpha) \exp(-TE/T2)$$
$$M_{z,1} = M_0 \times [1 - \exp(-TI/T1)]$$
$$E_1 = \exp(-TR/T1)$$
$$P_{eq} = M_0 \frac{1 - E_1}{1 - E_1 \cos \alpha}$$
$$P_{T1} = \frac{1 - (E_1 \cos \alpha)^N}{1 - E_1 \cos \alpha}$$

The link with concentration used the usual relaxivities equations:

$$R1 = 1/T1 = 1/(T1_0) + r_1 \times [C]$$
$$R2 = 1/T2 = 1/(T2_0) + r_2 \times [C]$$

To obtain this equation, we assumed a perfect saturation, followed by a series of SPGR pulses after a recovery time of TI. The process can be reconstructed following the methodology of (Hänicke et al., 1990). More detailed, step by step, approach is available in chapters 1.2.a. and 1.2.b. of (Rebbah, 2019).

| Parameter | Description |
|---|---|
| $M_0$ | Steady state magnetization |
| $T(1/2)_0$ | Initial $T(1/2)$ before bolus arrival |
| $r_{(1/2)}$ | relaxivities of $T(1/2)$ |
| $TI$ | Recovery time between saturation pulse and acquisition |
| $TR$ | Repetition time |
| $TE$ | Echo time |
| $\alpha$ | Flip angle |
| $N$ | Number of acquired line in k-space |

## Appendix C. Quantitative perfusion analysis

All the analysis use the dual saturation acquisition described in the in vivo database section. The quantitative perfusion analysis was performed using a deconvolution algorithm with a free-form residue function (Olea Medical).

### C.1. Main myocardial signals

The analyze were performed using the different AIFs and the 5 main myocardial signals. The results were evaluated through:

- Comparison with the extracted main signals: MSE and R2 of the estimated reconstructed myocardial signals

- Comparison with the residue function derived from $\text{AIF}_{unsat}$: MSE of the estimated residue function with $\text{AIF}_{sat}$ and $\text{AIF}_{db}$

- Comparison with the residue function derived from $\text{AIF}_{unsat}$: relative error of the estimated MBF using $\text{AIF}_{sat}$ and $\text{AIF}_{db}$

All results are depicted in Figure 5. It is observed that the estimation achieved with $\text{AIF}_{db}$ demonstrates greater concordance with $\text{AIF}_{unsat}$ compared to $\text{AIF}_{sat}$. Specifically, there is a factor 10 improvement favoring $\text{AIF}_{db}$ in terms of relative error of the estimated MBF.

### C.2. Mapping results

We selected one volunteer from the in vivo database to present the mapping results that are gathered in Figure 6.

The mapping results reinforce the earlier observations, indicating stronger agreement between $\text{AIF}_{unsat}$ and $\text{AIF}_{db}$ than between $\text{AIF}_{unsat}$ and $\text{AIF}_{sat}$. Notably, the error associated with $\text{AIF}_{db}$ is lower compared to the findings observed for the main myocardial signals dataset. This variance might stem from the extraction procedure of the main signals, potentially introducing erroneous signals through the averaging of divergent signals within the same cluster. This aspect warrants further investigation.

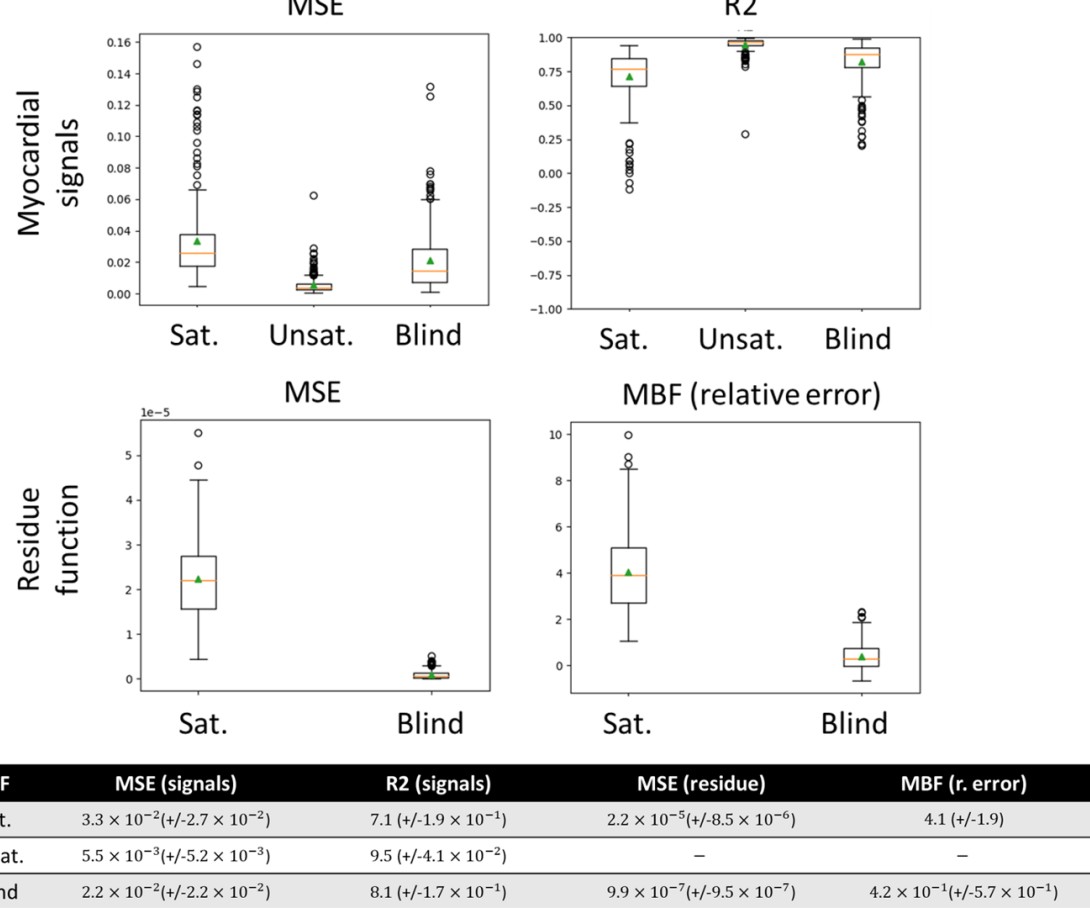

| AIF | MSE (signals) | R2 (signals) | MSE (residue) | MBF (r. error) |
|---|---|---|---|---|
| Sat. | $3.3 \times 10^{-2}(+/-2.7 \times 10^{-2})$ | $7.1 (+/-1.9 \times 10^{-1})$ | $2.2 \times 10^{-5}(+/-8.5 \times 10^{-6})$ | $4.1 (+/-1.9)$ |
| Unsat. | $5.5 \times 10^{-3}(+/-5.2 \times 10^{-3})$ | $9.5 (+/-4.1 \times 10^{-2})$ | — | — |
| Blind | $2.2 \times 10^{-2}(+/-2.2 \times 10^{-2})$ | $8.1 (+/-1.7 \times 10^{-1})$ | $9.9 \times 10^{-7}(+/-9.5 \times 10^{-7})$ | $4.2 \times 10^{-1}(+/-5.7 \times 10^{-1})$ |

Figure 5: Quantification of the extracted in vivo main myocardial signals obtained with $\text{AIF}_{unsat}$, $\text{AIF}_{sat}$ and $\text{AIF}_{db}$. The results are presented in term of MSE, R2 and relative error of the estimated MBF. The table presents the average and the standard deviation of the results.

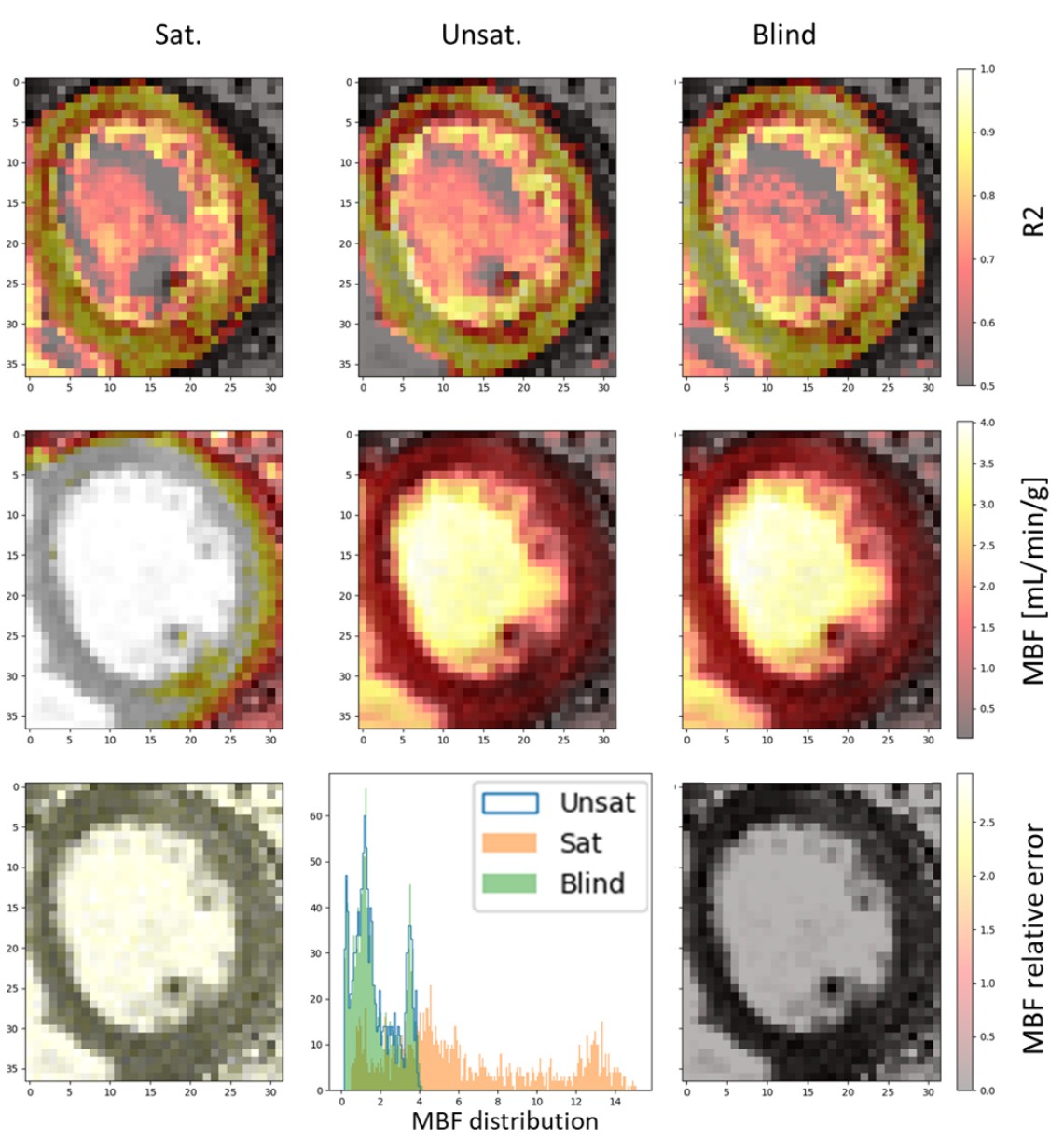

Figure 6: Mapping results: quantification of MBF using $AIF_{unsat}$, $AIF_{sat}$ and $AIF_{db}$. $AIF_{sat}$ and $AIF_{db}$ are compared to $AIF_{unsat}$ in term of relative error of MBF (last row), and the distributions of MBF in the selected bounding box are plotted for each AIF. R2 was computed between the acquired and estimated signals.

# Appendix D. Building model and comparison of training strategies

## D.1. Full blind deconvolution model

Theoretically, the blind deconvolution problem could be performed solely using the main signals. We propose here to compare the results of the full blind deconvolution model with the proposed model. The full blind deconvolution model is constructed using the same architecture as the proposed model, except that the layers associated with $AIF_{sat}$ are omitted, utilizing only the main signals as input. The model is trained using the same simulated database as the proposed model. Dropout layers with a rate of 0.2 are incorporated to prevent zero values in the output. Both models are trained with a maximum of 150 epochs, with a stopping criterion based on the loss of the validation dataset.

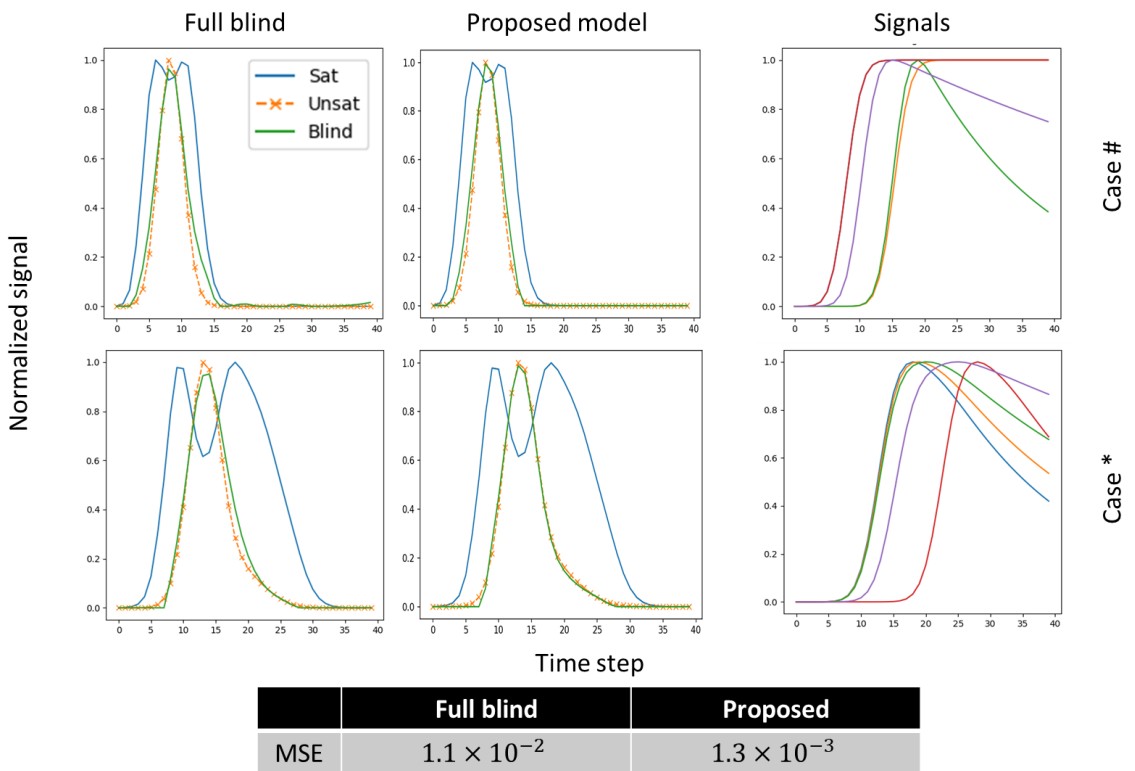

| | **Full blind** | **Proposed** |
|---|---|---|
| MSE | $1.1 \times 10^{-2}$ | $1.3 \times 10^{-3}$ |

Figure 7: Comparison between the proposed model and the full blind deconvolution model. The MSE is computed using the validation dataset.

The results are presented in Figure 7. Notably, we observe a higher MSE for the full blind deconvolution model compared to our proposed model. Such behavior is anticipated, as the input signals may exhibit limitations such as low variability or significant delay. Introducing an initial $AIF_{sat}$ constraint in our proposed model helps mitigate these limitations.

In (Choi et al., 2020) and (Scannell et al., 2023), the authors advocated for using only the $AIF_{unsat}$ as input to train the network, demonstrating the feasibility of such an ap-

proach. However, these results are likely influenced by the limited dataset variability, as these studies utilized perfusion acquisitions from a few centers with restricted protocol parameter variations. Additional inputs such as the main signals or the $\text{AIF}_{sat}$ can provide constraints to enhance the solution. One could also experiment with fewer primary signals, theoretically achieving accurate $\text{AIF}_{unsat}$ estimation. Nevertheless, such approaches will expose the process to a higher risk of error.

### D.2. Comparison of training strategies

In this section, we compare the same proposed model (with dropout rate of 0.2) trained with:

- The simulated database as described in the main text

- The dual saturation acquisitions from the in vivo database

- The dual saturation acquisitions from the in vivo database to fine tune the model trained with the simulated database

The performances of the models were evaluated using 10 cases from in vivo database and 10,000 cases of the simulation training dataset. To train with in vivo data, we split them in two sets: 26 for training and 7 for testing and managing the training process. The performances were evaluated in terms of MSE and R2. The results are presented in Figure 8.

The results highlight the advantage of the models trained with the simulated database. The fine-tuned model exhibits a slight improvement in terms of MSE and R2 compared to the model trained solely with the in vivo database. However, the model trained with the simulated database outperforms the fine-tuned model for in vivo and simulation dataset. These results lend support to the proposed strategy and its ability to generalize.

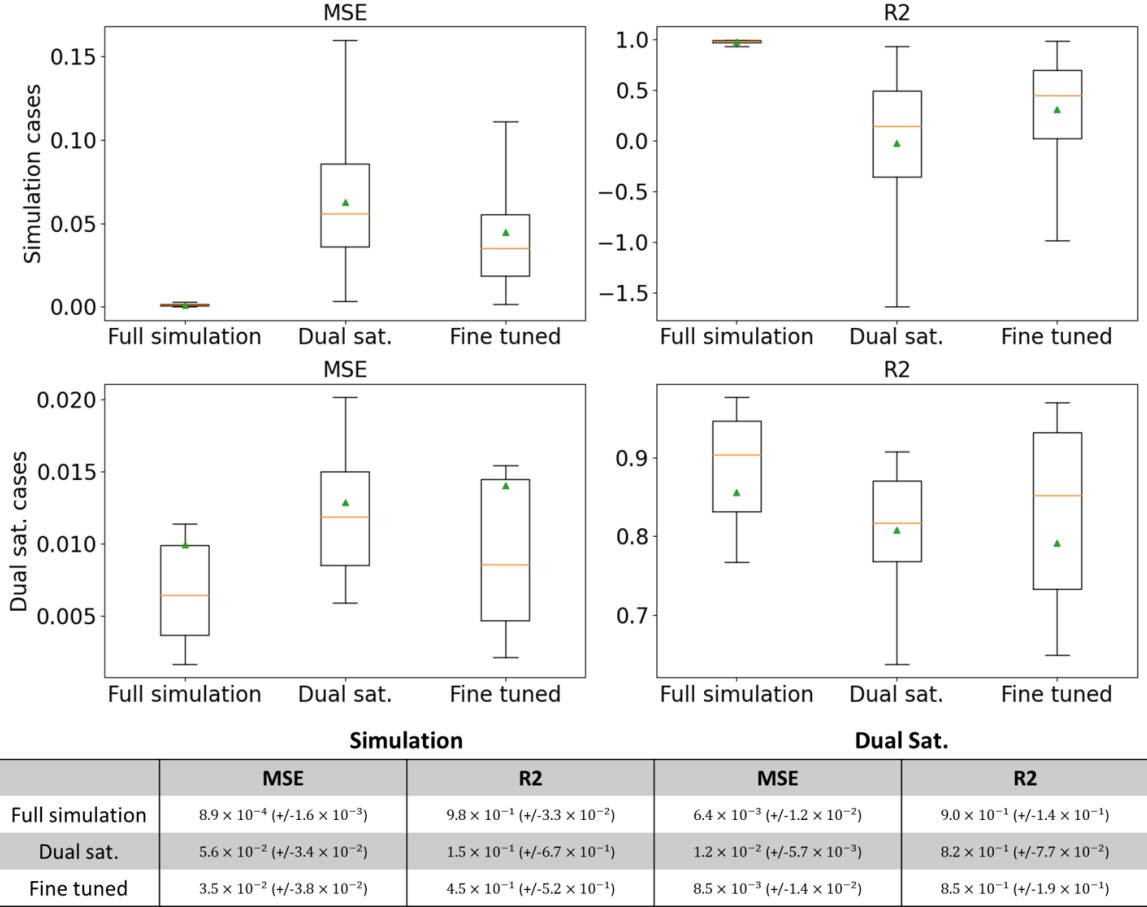

| | Simulation | | Dual Sat. | |
|---|---|---|---|---|
| | **MSE** | **R2** | **MSE** | **R2** |
| Full simulation | $8.9 \times 10^{-4}$ (+/-$1.6 \times 10^{-3}$) | $9.8 \times 10^{-1}$ (+/-$3.3 \times 10^{-2}$) | $6.4 \times 10^{-3}$ (+/-$1.2 \times 10^{-2}$) | $9.0 \times 10^{-1}$ (+/-$1.4 \times 10^{-1}$) |
| Dual sat. | $5.6 \times 10^{-2}$ (+/-$3.4 \times 10^{-2}$) | $1.5 \times 10^{-1}$ (+/-$6.7 \times 10^{-1}$) | $1.2 \times 10^{-2}$ (+/-$5.7 \times 10^{-3}$) | $8.2 \times 10^{-1}$ (+/-$7.7 \times 10^{-2}$) |
| Fine tuned | $3.5 \times 10^{-2}$ (+/-$3.8 \times 10^{-2}$) | $4.5 \times 10^{-1}$ (+/-$5.2 \times 10^{-1}$) | $8.5 \times 10^{-3}$ (+/-$1.4 \times 10^{-2}$) | $8.5 \times 10^{-1}$ (+/-$1.9 \times 10^{-1}$) |

Figure 8: Comparison of the proposed model trained with different datasets. The fine-tuned approach is initially trained with the simulated database and then fine-tuned with the in vivo database. MSE and R2 are computed using subsets of the in vivo (last row) and testing simulation (upper row) datasets.

## Appendix E. Analyze of zeros outputs

The proposed model exhibits a peculiar phenomenon where certain temporal positions of the corrected AIF consistently display zero values in each training iteration. These specific temporal positions vary with each new training seed, which is a variation of the randomly selected cases for training. The cause of this behavior remains unclear; however, we propose to investigate the impact of the dropout layers on a such behavior.

The model was trained with different dropout rate (0 and 0.2) with two different seeds. The results are presented in Figure 9.

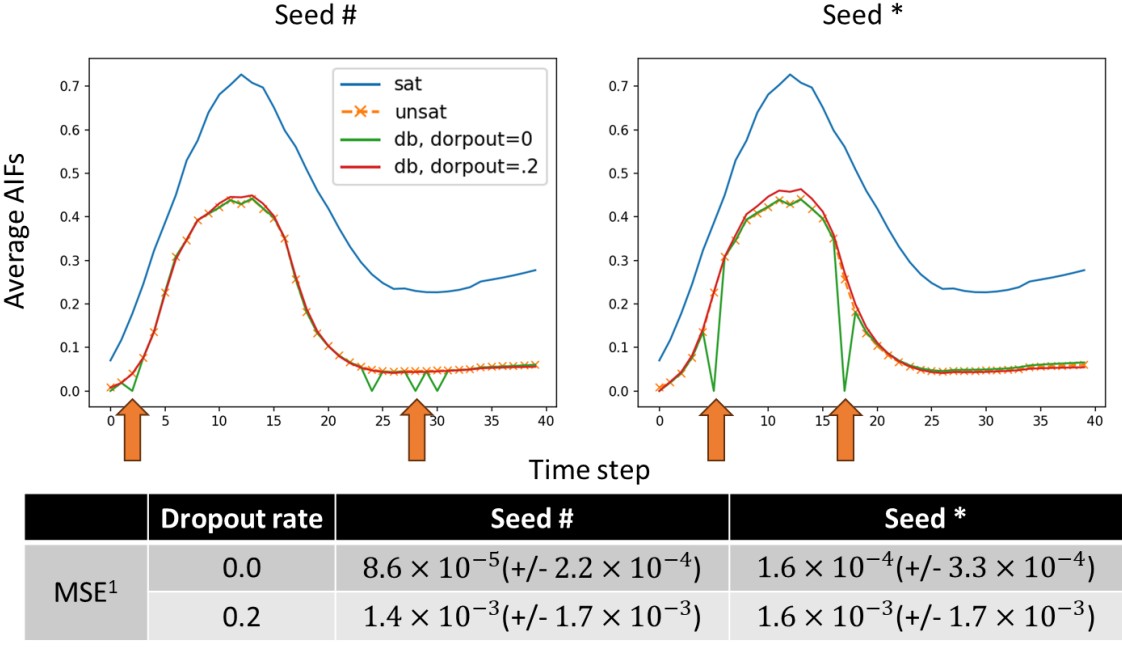

| | Dropout rate | Seed # | Seed * |
|---|---|---|---|
| MSE[1] | 0.0 | $8.6 \times 10^{-5}(+/- 2.2 \times 10^{-4})$ | $1.6 \times 10^{-4}(+/- 3.3 \times 10^{-4})$ |
| | 0.2 | $1.4 \times 10^{-3}(+/- 1.7 \times 10^{-3})$ | $1.6 \times 10^{-3}(+/- 1.7 \times 10^{-3})$ |

Figure 9: Impact of the dropout layers on the zero outputs. The sub-figures represent the average AIFs across the testing dataset for two models trained with different seeds. The orange arrows highlight some zeros points. 1: The positions with zero values are excluded from the MSE computation

The positions with zero values vary with the selected seeds, and the addition of dropout layers effectively addresses the issue. However, the MSE computed without considering the zero positions is higher for the case with dropout layers. Such results suggest that if a robust post-processing step is implemented to handle the zero points, the dropout layers could potentially be removed. In the algorithm presented in the main text, we proposed detecting these zero positions and interpolating the values at those time points.

