# OpenReview forum: "Deep blind arterial input function: signal correction in perfusion cardiac magnetic resonance"
_MIDL.io/2024/Conference — MIDL 2024 Poster_

### Official Review · Reviewer_Ftwb · 2024-02-25

**Confidence:** 4
**Preliminary Rating:** 2
**Final Rating:** 2.5

**Summary:**

The authors propose a CNN model trained on simulated data to correct the arterial input function signal in perfusion cardiac MR.

**Strengths:**

The paper tackles an important problem, and the proposed CNN solution leverages simulated data for training, which is important given the scarcity of data in the field. Another strength of the paper is the simplicity of the proposed method.

**Weaknesses:**

In my opinion, the main weaknesses of the paper are the lack of comparisons against other methods in the literature and the fact that the corrected AIF in the experiments consistently exhibited values of zero. Still, the authors were not able to explain why this happened.

I believe the authors also missed the opportunity to compare their proposed model against a fine-tuned version of their model using in vivo data, which would increase our understanding of the discrepancy between training in simulated data and in vivo data.

**Detailed Comments:**

Minor comments:

- Why the MSE arrow is entering the CNN block in Figure 1?
-  Was training on nearly 4M simulated data points necessary for the proposed model to achieve its performance?

**Justification Of Final Rating:**

Based on the authors' responses, I decided to raise my evaluation to borderline reject. However, I still think the work is not ready for publication. Understanding why  AIF in the experiments consistently exhibited values of zero is important. Is this a finding or a bug/mistake in their code?

**Justification Of The Preliminary Rating:**

In my opinion, the paper would benefit from adding more comparisons (literature and training or fine-tuning using in vivo data)  and investigating the peculiar results that the model seems to be producing.

**Questions To Address In The Rebuttal:**

- Why the corrected AIF in the experiments consistently exhibited values of zero?
- How does the proposed model compare against others in the literarture?
- How does the proposed model compare against training using in vivo data and training on simulated data and fine-tuning on in vivo data?

---

> ### Author Response · Authors · 2024-03-18
> **Response**
>
> We thank the reviewer for its insightful feedback. We hope that the new version of the article addresses all its questions.
>
>
> •	Why the MSE arrow is entering the CNN block in Figure 1?
>
> To represents the gradient backpropagation (based on MSE).
>
> •	Was training on nearly 4M simulated data points necessary for the proposed model to achieve its performance?
>
> Indeed, the actual number itself may not be as crucial as the coverage of the possibility space. Testing a broader sampling (thus fewer cases) but ensuring at least the same coverage of the space (or even greater coverage) could be an interesting point to investigate.
>
> •	Why the corrected AIF in the experiments consistently exhibited values of zero?
>
> For the moment, the reason is not clear. However, we have added an appendix (E) to discuss this point in more detail. It appears that certain nodes are not "used", and dropout forces them to be involved. We hope to gain insights and potential avenues for further investigation during the conference.
>
> •	How does the proposed model compare against others in the literarture?
>
> The comparison with deep learning methods is in Appendix D.2. Comparing with other non-deep learning methods seems more complex to implement. We chose the "dual saturation" method as the gold standard. This leaves: the population AIF method and direct signal correction. Testing the former seems unviable as its limitations have long been documented. Comparing with the latter is more debatable: either the sequence modeling is near-perfect and the approach result is almost error-free, or the modeling is inadequate and claiming our approach is better would be of little interest. We prefer to emphasize one of the primary benefit of our approach: the "user-friendly" aspect. No modeling is required, and no information other than image intensities is used.
>
> •	How does the proposed model compare against training using in vivo data and training on simulated data and fine-tuning on in vivo data?
>
> This point is added in annexe D.

---

### Official Review · Reviewer_noxE · 2024-02-26

**Confidence:** 3
**Preliminary Rating:** 2
**Recommendation:** Poster
**Final Rating:** 4

**Summary:**

The authors proposed a deep learning methods to get corrected AIF from saturated AIF and a set of myocardial tissue signals, using a large of synthetic dataset for network training. The trained network was evaluated using a dual-saturated sequence to compare the corrected AIF with the unsaturated version. The result showed that the proposed network generated corrected AIF with better R2 scores than the saturated AIF.

**Strengths:**

The paper is well-written and easy to understand. The authors gave a comprehensive detailed background introduction and showed the challenges in the AIF estimation. Due to the difficulties of acquiring large data for network training, the authors proposed to generate simulated data based on Parker’s simplified form and MRI Bloch equation simulation. This is very good since it is hard or impossible to get real data for network training. In addition, the proposed network combined the information from acquired saturated AIF and 5 main signals from perfused tissues, rather than directly using acquired saturated AIF to get unsaturated AIF, which might better perform the unsaturated AIF estimation.

**Weaknesses:**

The proposed methods utilized simulated data for training, then the trained network was applied on in-vivo data. The difference between the simulated data and in-vivo data could cause performance degradation. The authors failed to take into this consideration. Therefore, the authors's claiming that "Training the model with simulated data allows to easily explore perfusion parameters, enhancing its robustness and generalizability compared to strategies using limited number of in-vivo data to train the network." is not convincing. The author should compare the proposed methods with 1) simulated data for pre-training, then apply transfer learning on in-vivo data. 2) in-vivo data only for network training. In addition, the authors should compare the proposed methods with conventional non-deep learning methods and existing deep learning methods, to have a comprehensive performance evaluation. Regarding to the network architecture, since the inputs include acquired saturated AIF and 5 main signals from perfused tissues, it is better to have an ablation study to compare several networks with different inputs, such as 1) only saturated AIF. 2) saturated AIF and 1-5 main signals from perfused tissues. Since the AIF is time series signal, it is better to compare the the proposed network architecture with some other network architectures such as RNNs. Last one, the authors observed that certain temporal positions of the corrected AIF consistently exhibited values of zero, while the cause is unclear. Therefore, it is better to double-check the simulation procedure. It is better to exclude these outliers in the training data to affect the training even though it might be tiny.

**Detailed Comments:**

More experiments are needed in this paper.
1. Ablation study about the training data. 1) simulated data only for network training. 2) simulated data for pre-training, then apply transfer learning on in-vivo data. 3) in-vivo data only for network training.

2. Ablation study about the network input. 1) only saturated AIF. 2) saturated AIF and 1-5 main signals from perfused tissues.

3. Compare the proposed method with conventional methods and other existing deep learning methods.

4.  About corrected AIF consistently exhibited values of zero, the authors should find out the reason.

**Justification Of Final Rating:**

Thanks for the authors addressing my questions. The authors provided more experiments in the revision which makes the paper more solid. Based on the revision provided, I would like to increase my final rating.

**Justification Of The Preliminary Rating:**

Based on my best, I acknowledged that the authors proposed a deep learning methods to address the challenging problem in AIF. However, the paper lacks necessary experiments and ablation studies, to demonstrate their methods indeed outperform others.

**Questions To Address In The Rebuttal:**

1. The author adopted Parkers' simplified from to generate simulated database. Is there sophisticated model to do simulation and how do you think it might affect the result?
2.  The authors mentioned that time step units rather than absolute time length in generating simulated data for training. Did the authors do data interpolations when applied to the in-vivo data which could be acquired with different time resolution?
3. Does the corrected AIF is used for quantitative evaluation of perfusion imaging? If so, could the author evaluate it the estimated corrected AIF besides MSE and R2?

**Special Issue:**

No

---

> ### Author Response · Authors · 2024-03-18
> **Response**
>
> The authors thank the reviewer for the numerous suggestions for improvement that we have incorporated into the new version of the article.
>
> 1.	Ablation study about the training data. 1) simulated data only for network training. 2) simulated data for pre-training, then apply transfer learning on in-vivo data. 3) in-vivo data only for network training.
>
> Added in Appendix D.2.
>
> 2.	Ablation study about the network input. 1) only saturated AIF. 2) saturated AIF and 1-5 main signals from perfused tissues.
>
> Added in Appendix D.1. Additionally, we mention that direct comparison may not cover all scenarios. Blind deconvolution, for instance, theoretically could lead to correct estimations with only 2 signals. However, even if tests with our data had shown this, the obtained model would have been highly sensitive to errors. Having 5 signals + an initial AIF estimation helps guard against potential errors.
>
> 3.	Compare the proposed method with conventional methods and other existing deep learning methods.
>
> The comparison with deep learning methods is in Appendix D.2. Comparing with other non-deep learning methods seems more complex to implement. We chose the "dual saturation" method as the gold standard. This leaves: the population AIF method and direct signal correction. Testing the former seems unviable as its limitations have long been documented. Comparing with the latter is more debatable: either the sequence modeling is near-perfect and the approach result is almost error-free, or the modeling is inadequate and claiming our approach is better would be of little interest. We prefer to emphasize one of the primary benefit of our approach: the "user-friendly" aspect. No modeling is required, and no information other than image intensities is used.
>
> 4.	About corrected AIF consistently exhibited values of zero, the authors should find out the reason.
>
> Appendix E is added to detail this point.
>
> 1.	The author adopted Parkers' simplified from to generate simulated database. Is there sophisticated model to do simulation and how do you think it might affect the result?
>
> Yes, we could have used gamma functions instead of Gaussians, often described as closer to reality. Additionally, we modeled two passages of the contrast agent; more passages could have been added. But more importantly, we could have also modeled dual-bolus profiles. It seems that the gains from the former approaches may be limited. However, simulating dual-bolus profiles could indeed greatly improve the generalization of the model.
>
> 2.	The authors mentioned that time step units rather than absolute time length in generating simulated data for training. Did the authors do data interpolations when applied to the in-vivo data which could be acquired with different time resolution?
>
> Thank you for pointing out this omission. Yes, linear interpolation is used if necessary. A sentence has been added to the methodology section.
>
> 3.	Does the corrected AIF is used for quantitative evaluation of perfusion imaging? If so, could the author evaluate it the estimated corrected AIF besides MSE and R2?
>
> The overall clinical evaluation is ongoing. However, we have added the MBF calculated with the 5 main myocardial signals extracted. A comparison is proposed in addition to R2 and MSE in the main text, and a more detailed version is in Appendix C. Additionally, mapping results for one volunteer have also been added in Appendix C.

---

### Official Review · Reviewer_GJJE · 2024-03-05

**Confidence:** 3
**Preliminary Rating:** 3

**Summary:**

The authors proposed to use a CNN with the saturated AIF and a set of myocardial tissue signals as inputs, to generate the
corrected AIF as the output. A simulated datasets was created to train the network, and a real clinical dataset was used for testing. The results show that the proposed network can correct the acquisition-induced effects on the AIF and also displays the ability for generalization.

**Strengths:**

The application of this paper was well explained and the authors have comprehensively included the previous research regarding this application. The results of the proposed method are interesting, especially the generalization to real clinical data by training only on the simulated data.

**Weaknesses:**

- It seems not clear to me why 2 branches of the CNN are 1D and 2D for different inputs. It would be helpful to explain the motivation/reason for this design.

- The zero point issues in Figure 2 may need more discussion. During training, do the targets in the simulated database have the patterns of such zero points? If not, this issue is likely to be caused by the implementation.

**Detailed Comments:**

grammar errors:
last sentence of the introduction: to returns --> to return
page 6: The results are presented in Figure 3 and Figure 4 presents some selected cases.

The motivation of the network design and the major findings (e.g., section 3.2) need to be made more clearer.

**Justification Of The Preliminary Rating:**

The current version may need more clarifications. It would be more clear to the readers if the motivation of the proposed method and the major findings can be further clarified. For example, at the end of the introduction, it would be helpful to highlight the difference between the proposed method and the previous methods.

**Questions To Address In The Rebuttal:**

It would be more clear to the readers if the motivation of the proposed method and the major findings can be further clarified. For example, at the end of the introduction, it would be helpful to highlight the difference between the proposed method and the previous methods.

---

> ### Author Response · Authors · 2024-03-18
> **Response**
>
> Firstly, the authors express their gratitude to the reviewer for its interest in this work. We have submitted a revised version of the article, which we hope addresses all of the requests.
>
> - "It seems not clear to me why 2 branches of the CNN are 1D and 2D for different inputs. It would be helpful to explain the motivation/reason for this design."
>
> The reason lies in the shape of the two inputs: 1D for the AIF and 2D for the signals (5 main signals).
>
> - "The zero point issues in Figure 2 may need more discussion. During training, do the targets in the simulated database have the patterns of such zero points? If not, this issue is likely to be caused by the implementation."
>
> Indeed, we have added an appendix (E) dedicated to this issue. As you will notice, the pattern of zeros is not present in the database. Furthermore, the same implementation leads to their appearance at different points depending on the training performed. Finally, the same implementation can entirely remove them by adding dropout layers. It seems that dropout layers 'force' the network to involve all nodes. Why this is not the case without them is currently unclear. This is one of the reasons motivating us to submit to a conference dedicated to medical AI topics. We hope that this point will raise the interest of participants as it did for all the reviewers.
>
> - "It would be clearer to the readers if the motivation of the proposed method and the major findings can be further clarified. For example, at the end of the introduction, it would be helpful to highlight the difference between the proposed method and the previous methods."
>
> Indeed, thank you for pointing it out. A sentence has been added at the end of the introduction for this purpose.

---

### Meta-Review · Area_Chair_BK1r · 2024-04-05

**Recommendation:** Accept (Poster)
**Confidence:** 4

**Metareview:**

The authors proposed a deep learning method trained on simulated data to correct the saturated arterial input function (AIF) signal in perfusion cardiac MR.  This is a real and clinically significant problem. Given the saturated AIF and a set of myocardial tissue signals, the network predict the unsaturated AIF. The trained network was evaluated using a dual-saturated sequence to compare the corrected AIF with the unsaturated version. The results show some improvement and the proposed network generated corrected AIF with better R2 scores than the saturated AIF.

---

### Decision · Program_Chairs · 2024-04-05

Accept (Poster)